# Modeling strict age-targeted mitigation strategies for COVID-19

**Maria Chikina**[1]ᴑ*, **Wesley Pegden**[2]ᴑ

**1** Department of Computation and Systems Biology, University of Pittsburgh, Pittsburgh, PA, United Status of America, **2** Department of Mathematical Sciences, Carnegie Mellon University, Pittsburgh, PA, United Status of America

ᴑ These authors contributed equally to this work.
* mchikina@pitt.edu

## Abstract

We use a simple SIR-like epidemic model integrating known age-contact patterns for the United States to model the effect of age-targeted mitigation strategies for a COVID-19-like epidemic. We find that, among strategies which end with population immunity, strict age-targeted mitigation strategies have the potential to greatly reduce mortalities and ICU utilization for natural parameter choices.

## 1 Introduction

In this paper, we use a simple age-sensitive SIR (Susceptible, Infected, Removed) model integrating known age-interaction contact patterns to examine the potential effects of age-heterogeneous mitigations on an epidemic in a COVID-19-like parameter regime. Our goal is to demonstrate the qualitative point that for an epidemic with COVID-19-like parameters, age-sensitive mitigations can result in considerably less mortality and ICU usage than homogeneous mitigations, among strategies which end with population immunity.

We model mitigations which result in a 70% reduction in transmission rates among all of the population except for a relaxed group. In our models, natural transmission rates correspond to an $R_0$ of 2.7. (In S3 Section in S1 File, we consider other parameter choices). We consider relaxation strategies based on age thresholds; for example, relaxing restrictions on all people under 50. Because people of different ages often live in the same household, and because other risk factors might also be used to inform mitigation efforts, we consider strategies where only some fraction of people under the age threshold are subject to relaxed restrictions.

Note that while it is not at all surprising that targeting mitigations at higher risk groups can greatly reduce mortality during mitigations, our main contribution is to emphasize that this result still holds even if transmission rates for all groups are eventually relaxed. In particular, as we discuss in Section 2, the fact that this holds for COVID-19 depends on the coincidence that age-specific mortality rates for COVID are very strongly anti-correlated with age-specific contact patterns during periods of normal social interaction.

Related work in the context of MERS can be found in [1], and for H1N1 swine flu in [2].

**Data Availability Statement:** The code and data used can be downloaded from GitHub: https://github.com/mchikina/SIRageStratified.

**Funding:** Maria Chikina received no specific funding for this work. Wesley Pegden is funded by the National Science Foundation (DMS-1700365),

which in the Mathematical Sciences is expected to be acknowledged in all research projects done during the funding period, through a statement of the form \supported by NSF DMS-1700365", or something similar. Funding statement: The National Science Foundation played no role in study design, data col- lection and analysis, decision to publish, or preparation of the manuscript.

**Competing interests:** The authors have declared that no competing interests exist.

## 2 Motivation

It seems intuitively obvious that preferentially targeting at-risk populations for mitigations may reduce disease mortality by a large factor. Our goal in this paper is to show a more subtle (but still simple) qualitative effect; namely, that even if transmission rates return to normal in the future and the epidemic ends only when population immunity is sufficient to survive rein-troduction of infection, mitigations which were age-targeted can still achieve a large mortality reduction. For this question, the extent to which mortality can be greatly reduced by targeted mitigations depends heavily on the interaction between mortality rates and contact patterns. If the population most at risk of death from infection can sustain an epidemic without any role played by the younger population, then preferentially targeting mitigations at the at-risk popu-lation may accomplish relatively little, since when they are eventually lifted, the at-risk popu-lation will suffer high levels of infection anyways.

To examine this interplay between mortality rates and contact patterns, our model inte-grates the contact matrix generated for the United States in [3], which we process as described in S1 Section in S1 File. After preprocessing, this contact matrix, depicted on the left in Fig 1, represents the relative likelihood of reported interactions between pairs of age groups, after correcting for relative population sizes. (As a general rule, for example, one can see that the youngest age groups engage on average in many more interactions, especially with each other, than the oldest age groups. One can also see a square of elevated interactions corresponding to the workforce age-range).

A basic consequence of these contact patterns is that for a disease spread by contact whose patterns are captured by the methods of [3], and with an effective $R_0$ in moderate range (say 2 −3) the older population cannot generally sustain an epidemic on its own, while, on the other hand, the younger population is capable of sustaining an epidemic even in a hypothetical situa-tion where the older population is completely immune. This is illustrated on the righthand side of Fig 1; the upper-right panel shows the effective $R_0$ of an epidemic where only people

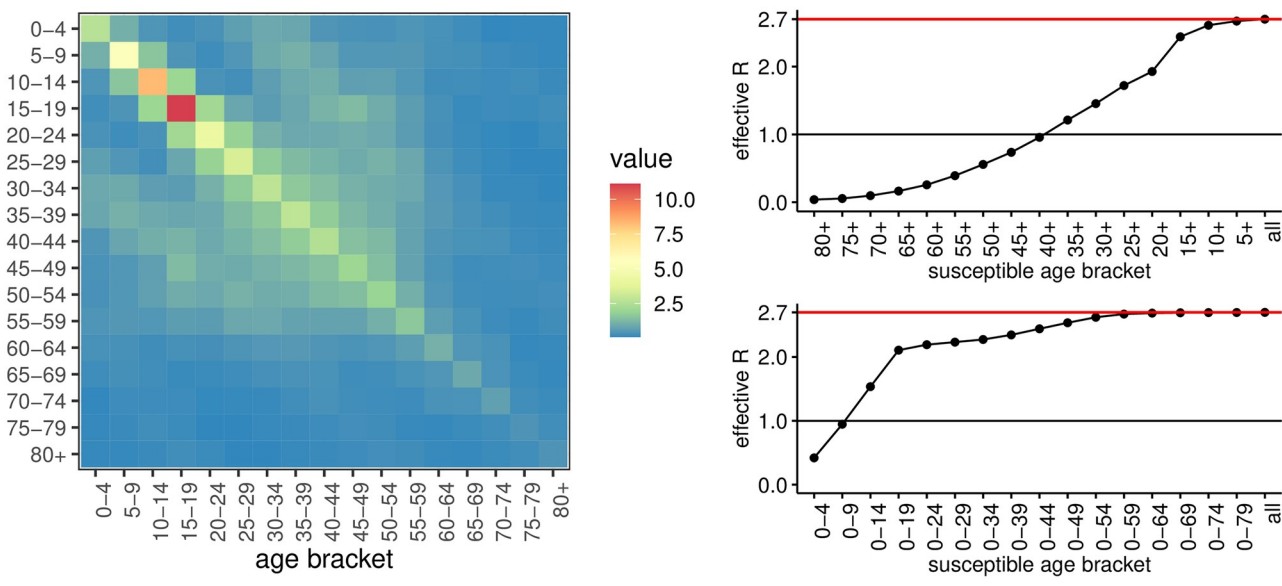

**Fig 1.** **Left**: The contact matrix we use, derived from the contact matrix generated in [3] for the United States. **Right**: Basic reproduction of an epidemic restricted to top- and bottom-segments of population by age, respectively, where the entire population corresponds to an $R_0 = 2.7$. *Note*: these figures assume young children pose a transmission risk equal to that of adults. We show corresponding figures where children pose a lower transmission risk in Section 6.

**Table 1. Mortality/ICU rates by age.**

| Age-group | Hospitalization rate | ICU rate for hospitalized cases | IFR |
|---|---|---|---|
| 0 to 9 | 0.1% | 5.0% | 0.002% |
| 10 to 19 | 0.3% | 5.0% | 0.006% |
| 20 to 29 | 1.2% | 5.0% | 0.03% |
| 30 to 39 | 3.2% | 5.0% | 0.08% |
| 40 to 49 | 4.9% | 6.3% | 0.15% |
| 50 to 59 | 10.2% | 12.2% | 0.60% |
| 60 to 69 | 16.6% | 27.4% | 2.2% |
| 70 to 79 | 24.3% | 43.2% | 5.1% |
| 80+ | 27.3% | 70.9% | 9.3% |

above a certain age are susceptible, while the lower panel shows the effective $R_0$ of an epidemic where only people under a certain age are susceptible. We see that a relatively small number of young individuals suffice to achieve a large $R_0$, while it takes a very large bracket of older individuals to cross the epidemic threshold. This outsize role of the younger population in epidemic spread is one reason that targeting influenza vaccination programs at younger populations can reduce mortality among high-risk populations by more than can be achieved by vaccination programs targeted at the high-risk population itself [4].

A distinguishing feature of the COVID pandemic is that its high fatality correlates strongly with age. Between ages 10 and 80, for example, the mortality rate varies by three orders of magnitude (Table 1). Importantly, the direction of this correlation is the opposite of that between age and interaction patterns; the age groups who would be expected to be most responsible for sustaining an epidemic under normal interaction patterns are, themselves, the least at risk from dying of it.

The goal of this paper is to explore the implications of this interplay between contact patterns and total mortality from COVID-19, under the assumption that some approximation to normal interaction patterns eventually resumes (or at least, the *relative* frequency of contacts by age group eventually resumes) before an effective vaccine can be widely deployed. Under this assumption, the epidemic will only die out when the remaining susceptible population cannot sustain an epidemic. The upper-right panel of Fig 1 illustrates that it is in principle possible for the infected/immune population to consist of younger individuals, whose fatality rates are much lower. (This means, for example, that a vaccine with high effectiveness only in younger individuals should nevertheless be expected to have a large effect at the population level). In contrast, we see in the lower-right panel that a much larger fraction of the population would have to be immune if made up entirely of older individuals.

In this paper we explore the extent to which the age-distribution of the infected can be shifted by mitigation strategies which do not perfectly isolate or separate populations based on age, but simply target transmission reductions at older populations.

## 3 The model

We use a simple SIR-like model acting on $n$ age groups. Let $I(t)$, $S(t)$, $R(t)$, $M(t)$ be vector-valued functions of time, each vector of length $n$, where the $j$th coordinate of each of $I(t)$, $S(t)$, $R(t)$, and $M(t)$, respectively, denote the the number of people in the $j$th age group who are infected, susceptible, recovered *or* deceased, and deceased, respectively at time $t$.

We let $\mathcal{C}$ be the age-group contact matrix. This is a symmetric matrix describing the frequency of transmission events between age groups, relative to the products of the population

sizes. (In particular, even if the sizes of age groups vary, $\mathcal{C}$ would be a constant matrix if we assumed people were equally likely to have have transmission interactions regardless of age). The constant $\alpha$ is the total recovery/removal rate, and the vector $\boldsymbol{\delta} = (\delta_1, \ldots, \delta_n)$ gives the death rate for each group separately.

Given a vector $v$, we let $\mathcal{M}_v$ denote the diagonal matrix whose $i$th diagonal entry is the $i$th coordinate of $v$.

The basic dynamics of the model are captured by the following vector differential equations:

$$\frac{dI(t)}{dt} = I(t)\mathcal{C}\mathcal{M}_{S(t)} \cdot \frac{1}{m} - \alpha I(t) \tag{1}$$

$$\frac{dS(t)}{dt} = -I(t)\mathcal{C}\mathcal{M}_{S(t)} \cdot \frac{1}{m} \tag{2}$$

$$\frac{dR(t)}{dt} = \alpha \cdot I(t), \tag{3}$$

with deaths tracked by

$$M(t) = \boldsymbol{\delta} \odot R(t) \tag{4}$$

Here $\odot$ denotes coordinate-wise multiplication, and $m$ is the total population. (As discussed in Section 3.3, these dynamics are elaborated for the purpose of tracking ICU utilization).

We emphasize that by using a simple SIR model, we are disclaiming the goal of making precise predictions regarding, e.g., the timing of infection peaks under various scenarios. Instead our goal is to understand the impact of age-targeted strategies relative to homogeneous ones.

## 3.1 The role of the contact matrix

When modeling mitigation strategies, the contact matrix $\mathcal{C}$ will be modified. In the absence of mitigations, we could simply set $\mathcal{C}$ to be a scalar multiple of the all-1's matrix $J$ (with the scalar set to reproduce the desired $R_0$ value). More realistically, we can incorporate known patterns of inter-age-group interactions. To do this, we use contact matrices generated by [3]; these are are extrapolations based on contact patterns originally measured by recording contact patterns for a large sample of people in European countries [5]. In this manuscript, we use the contact matrices from [3] generated for the United States.

One feature of using a non-constant contact matrix $\mathcal{C}$ is that the steady state proportions of a growing epidemic are no longer uniform among the age-groups in the population. For example, if we model a situation where we begin with $10^5$ infections in a much larger population, we do not expect those to be uniformly distributed by age, but instead distributed in a way which depends on $\mathcal{C}$.

To understand the dependence on $\mathcal{C}$, let $\boldsymbol{\gamma}$ denote a vector of length $n$ whose proportions represent infected fractions of each age group in the early growth of an epidemic (i.e., in the period where $R_0$ is often measured in the field). We call this the *early-stable proportion* vector. We let $\boldsymbol{\rho} = (\rho_1, \ldots, \rho_n)$ be the vector giving the fraction of the total population within each age group. If the *relative* magnitudes of the coordinates of $\boldsymbol{\gamma}$ are in steady-state (that is, if $\boldsymbol{\gamma}$ will grow in all coordinates uniformly), this means from (1), that for some constant $K$ we have that

$$\boldsymbol{\gamma}\mathcal{C}\mathcal{M}_{\boldsymbol{\rho}} - \alpha\boldsymbol{\gamma} = K\boldsymbol{\gamma}$$

In particular, $\gamma$ is a positive eigenvector of the positive matrix $\mathcal{CM}_\rho$. Such an eigenvector exists and is unique, by the Perron–Frobenius theorem. We let $\lambda_1$ denote its eigenvalue, which is the (unique) largest eigenvalue of $\mathcal{CM}_\rho$.

To tie our model to reported estimates of $R_0$ for COVID-19, we wish to compute a scaling factor $\beta$ for the matrix $\mathcal{C}$ which gives rise to an exponential growth in $\sum_{j=1}^{n} I_j(t)$ which corresponds to the empirical $R_0$ value computed early in an epidemic (before a substantial fraction of the population is infected), given an infection distributed according to the early-stable proportion vector $\gamma$. For the vector $\gamma$, multiplication by $\mathcal{CM}_\rho$ is equivalent to multiplication by the eigenvalue $\lambda_1$, thus for an infected population with age distribution proportional to $\gamma$, and writing $\mathcal{I}$ for $\sum_{j=1}^{n} I_j$, the dynamics near $t = 0$ reduce (1) to a standard single population SIR model with the infected population governed by the equation

$$\frac{d\mathcal{I}(t)}{dt} = \frac{\lambda_1}{m}\mathcal{I}(t) - \alpha\mathcal{I}(t).$$

In particular, we see that $R_0 = \frac{\lambda_1}{\alpha}$, and thus the correct scaling for the contact matrix $\mathcal{C}$ to replicate the known $R_0$ value is that which ensures that

$$\lambda_1 = R_0 \cdot \alpha.$$

See [6] for an application of the Peron-Frobenius eigenvalue in a related context.

## 3.2 Parameter selection

We use the recovery rate $\alpha = 1/14$, corresponding to a 2-week recovery period. (While disease course may vary by age, a large-scale study in Israel found recovery periods between 13 and 15 days for all age groups [7]). The mortality rate and rate of ICU admissions per infection we use are shown in Table 1, and are taken from those used in Report 9 of the team at Imperial College London [8] (see their Table 1).

Note that we do not apply a correction for asymptomatic cases, which may make our analysis conservative (pessimistic). Comparable (but not identical) estimates may be found from other sources. For example, see [9], S2 Table in S1 File.

Weighting the age groups by their population sizes, these mortality rates would correspond to assuming an overall IFR (infection fatality rate) of roughly 1%. But because of the age-variation of the mortality rate, the IFR depends to a large extent on the expected age-profile of the infected population. In particular, we will see that the numbers in Table 1 would correspond to an IFR of roughly 1% for a uniformly interacting population, but that the IFR would drop to around .5% for a population interacting with interaction patterns whose relative frequencies by age are representative of the typical patterns captured by the interaction matrices in [3]. (Note that the IFR one would expect to correspond to the interaction patterns resulting from indiscriminate mitigations may lie between these two). A key point of our paper is that if mitigations can be preferentially targeted at high-risk age groups, the IFR may decrease even further, even if mitigations eventually end at a point when population immunity must survive reintroduction of the disease. Note that this observation is consistent with the fact that there is a wide variation in observed COVID-19 IFR over space and time, depending on the age-profile of the infected population. For example, as of June 4, Singapore reports 24 deaths and 23,904 *recovered* cases, suggesting an exceptionally low IFR below $\approx 0.1\%$ at this point, which is easily understood given the fact that their outbreak has so far been largely confined to the dormitories of migrant workers, with a different age profile than the general population [10].

Source code for our model can be found at the entry for this paper at http://math.cmu.edu/~wes/pub.html.

### 3.3 ICU dynamics

There has been considerable variation over time and space between the correspondence of COVID case severity and ICU utilization. For example, the fatality rates for COVID patients receiving ICU care with known resolution have ranged 26% for Lombardy, Italy [11], to 37%-48% for the state of Georgia [12], 50% for the State of California [13], to 78% for the New York City area [14]. In part this can reflect differences in care practices; e.g., whether patients requiring noninvasive ventilation are typically managed within or outside the ICU setting.

In the present manuscript, we do not aim to predict precise levels of ICU utilization, but instead use a coarse model of ICU utilization simply as a rough proxy for utilization of scarce healthcare resources from very severe cases, for the purpose of making relative comparisons between strategies. Motivated by the parameter choices in [15], we assume that individuals spend, on average, 10 days of their infection period in the ICU (this is a weighted average of the durations used in [15] for ICU patients who recover or die, respectively). Since the number of ICU patients is very small compared with the rest of the population, our results are insensitive to the precise model of transmission attached to ICU patients, but our modeling code removes ICU patients from the population transmitting according to contact patterns. The complete description of our model accounting for ICU admissions can be found in S4 Section in S1 File.

## 4 Results

Our simulations all begin from an initial infection affecting 100, 000 individuals (distributed proportional to the early-stable proportion vector) in an otherwise fully susceptible population of size roughly $3 \times 10^8$. In all of our scenarios, we assume that normal transmission levels are linearly resumed between the 9- and 15-month marks from the start of the simulation.

As a first example, Fig 2A models the epidemic in the absence of any mitigations at all. Fig 2B models the epidemic in the presence of strict mitigations which, as in all our scenarios, are gradually relaxed between the 9- and 12-month marks. Each figure shows the size of the infected population over time, and the ICU utilization over time, by age group. Each figure title indicates Total Mortalities (TM) and the resulting effective IFR for the scenario. Note the light gray shading in the figures modeling mitigations serve as a reminder that normal transmission is linearly resumed between the 9- and 15- month marks.

### 4.1 The power of mitigations and natural heterogeneity

In all our scenarios, we assume that transmission rates linearly resume between the 9 month and 15 month points.

Fig 3A shows the outcome of optimal homogeneous measures. That is, among all transmission reductions which could be applied equally to all age groups and then gradually resumed as we assume, this is the choice of transmission reduction which minimizes deaths. Transmission levels in this scenario are reduced by 40% for all age groups. Note that this results in a reduction of mortalities by nearly 50%.

Fig 3B shows the outcome of the same mitigations ignoring the role of natural contact patterns in the population. In particular, for this scenario we have assumed a counterfactual where the likelihood of two people of different ages interacting is determined just by the relative sizes of the populations of the age-groups; this corresponds to a constant $\mathcal{C}$ matrix. We see 2.5 times more mortalities in this scenario, demonstrating the power of natural population

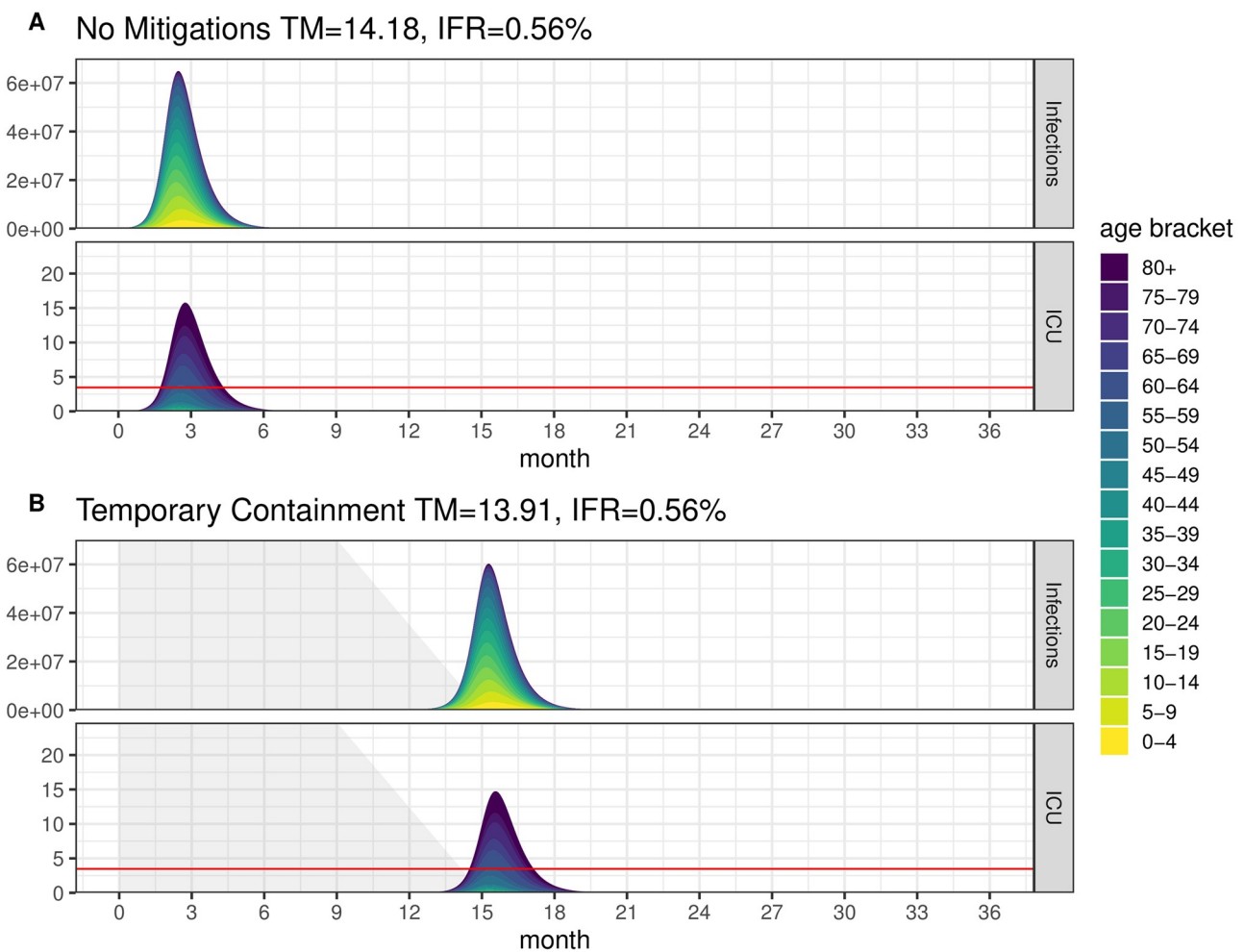

**Fig 2. A**: No mitigations. **B**: Containment followed by resumption in transmission levels between 9 and 15 months. (Scale for ICU figure is beds required per 10,000 people in population).

heterogeneity to reduce infection mortality for COVID-19. (The optimum choice for transmission reduction is roughly the same for the uniform contact matrix).

For each scenario, our figures show the resulting total mortalities (TM) and the effective IFR. Note that the power of age-targeted mitigations can be seen as coming from their potential to decrease this IFR by shifting the age-distribution of the infected population.

In all our figures, the scale ICU utilization is shown rescaled by a factor of $\frac{10{,}000}{\text{total population}}$. In other words, units for the ICU utilization figures is the ICU capacity—per 10,000 people—required to support ICU admissions at that level. The red line shows a nominal capacity level of 3.47 beds per 10,000 people.

## 4.2 Age-targeted mitigations

For our age-targeted mitigations, we consider relaxing mitigations just on those under 40, just on those under 50, and just on those under 60.

Since it is natural to expect targeted mitigations to be based on household-level of risks, because of cohabitation of younger and older adults, we consider, in each case, a scenario where only 2/3 of the younger population is subject to normal transmission levels.

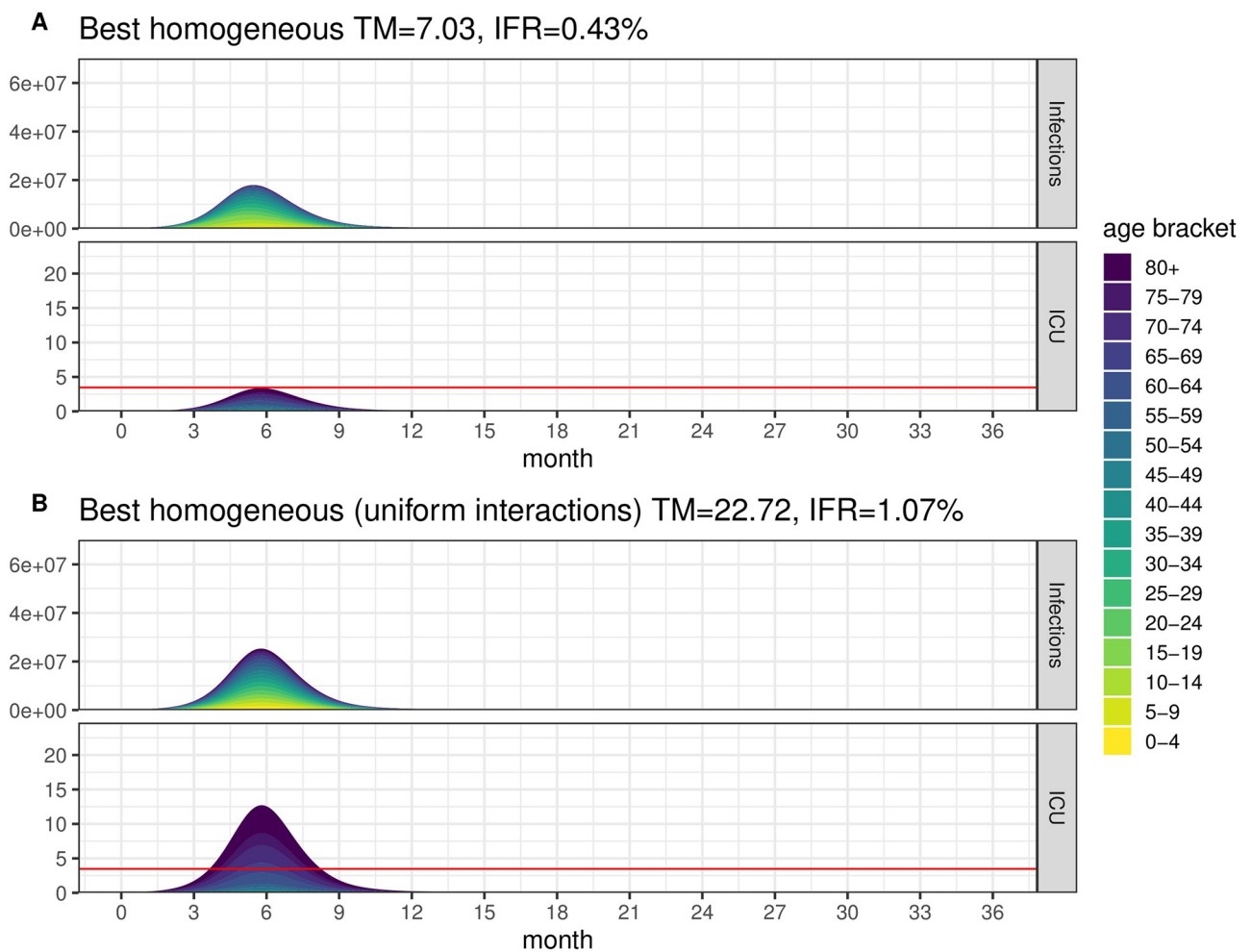

**Fig 3. A**: Optimum homogeneous mitigations with natural relative contact patterns. **B**: Homogeneous measures without the contact matrix. (Scale for ICU figure is beds required per 10,000 people in population).

In each of these scenarios, depicted in Figs 4, 5 and 6, we assume that the relaxed population is subject to normal transmission levels, while transmission to, from, and within the rest of the population is depressed by 70% from normal levels. (Results for other choices for these and other constants are discussed in S3 Section in S1 File).

What we see in Figs 4, 5 and 6 is that age-targeting has the potential to greatly reduce total mortalities compared with optimum choices for homogeneous measures. At the same time, we see that the best strategies for age-targeting are sensitive to the fraction of the younger population which can be released. In general, if too few people are released initially, a second wave occurs when transmission levels return to normal. Conversely, if too many are released, ICU utilization is high in the first wave. Thus the optimum choice for the age cutoff depends on the fraction of people we expect below the age cutoff to actually be released.

## 5 Age-specific mortality impact

It might be natural to suspect that age-specific strategies are simply trading mortalities in one group for mortalities in another. However, we find in our models that age-targeted restrictions can dramatically reduce mortalities among older populations with very small impacts on

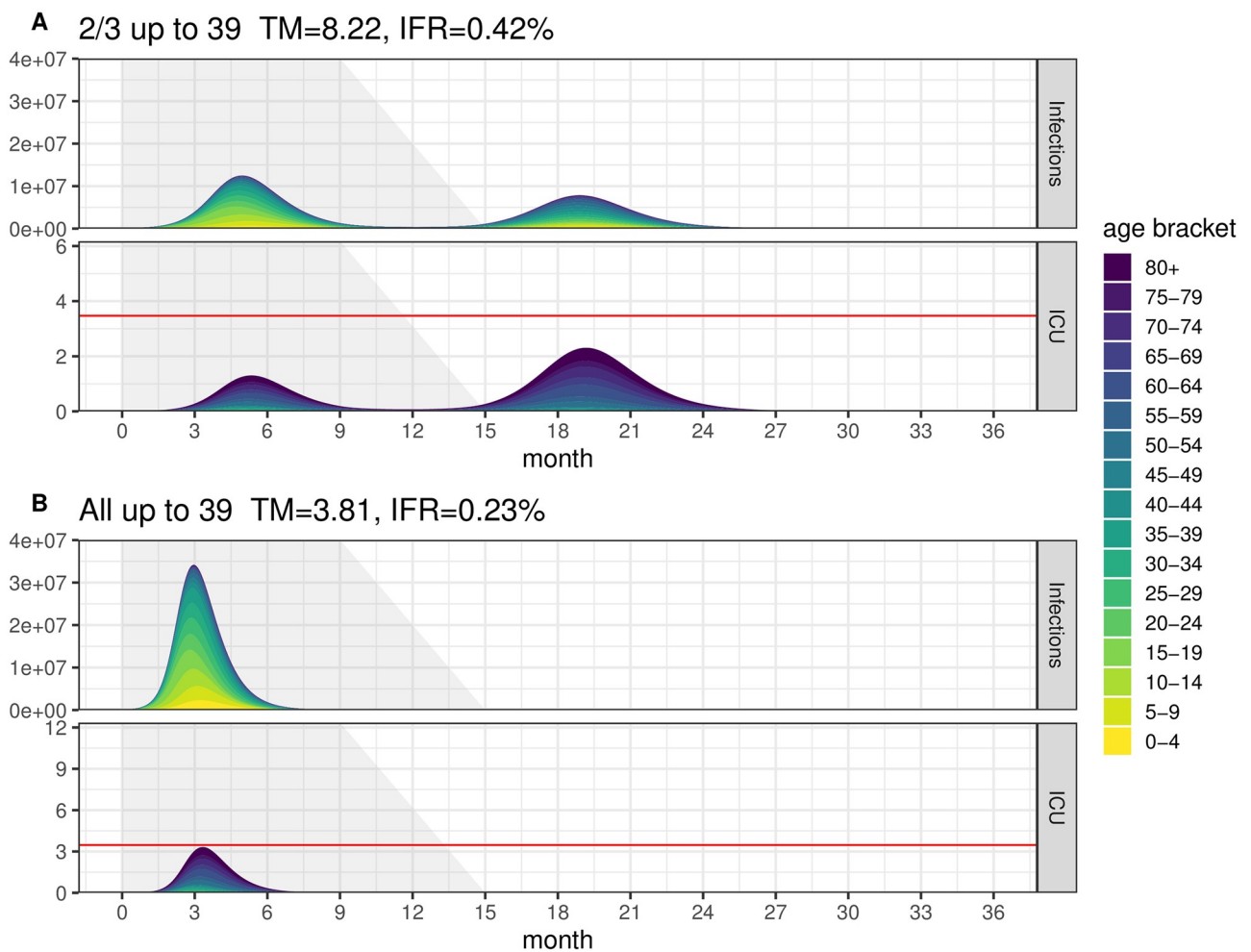

**Fig 4. A**: 2/3 under 39 released at normal levels. **B**: All under 39 released at normal levels.

mortality in younger populations; see Fig 7. In particular, because of the effect age-targeted measures can have on ICU over-utilization (whose impact on mortality we have not modeled), it is quite possible that well-calibrated age-targeted mitigations could improve typical outcomes for all age groups.

## 6 The effect of hypothetical differences in children

Some evidence has been presented that young children are less susceptible to infection from COVID-19 than adults; for example none of 234 tested children under 10 tested postive for COVID-19 in Vo, Italy, despite some living in households with infected members [16]. But evidence from different sources is mixed; for example, data from the UK has not shown a significant difference of infection rates in young children [17].

Separately, there is some evidence that even once infected, children are less infectious than adults: Countries tracking infection events by age (including, e.g., Iceland, where schools for young children have remained open) have seen very few events of young children infecting adults [18–20]. On the other hand, Jones et. al argue that epidemiological investigations are poorly suited to determine the transmission risk from children in the current environment [21].

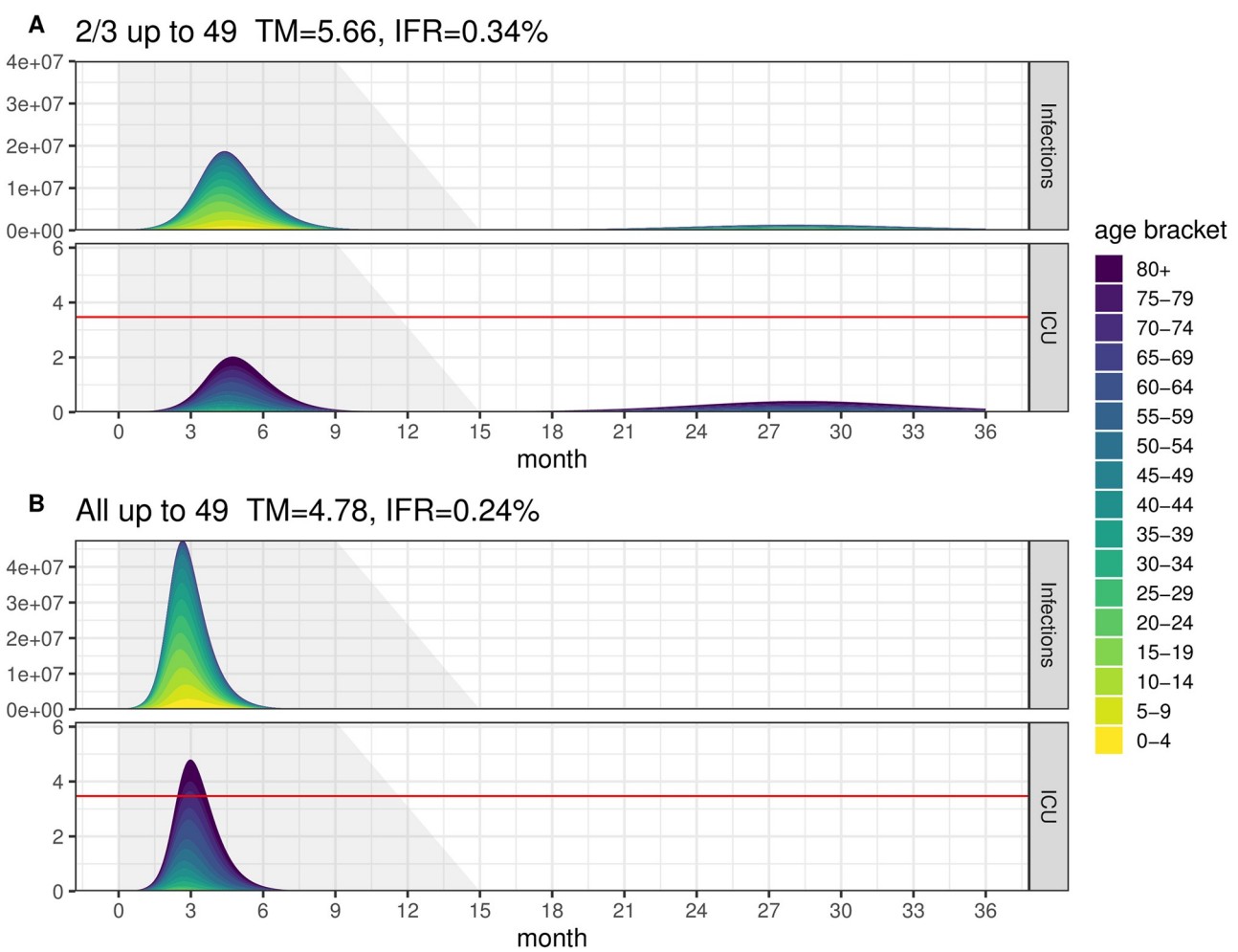

**Fig 5. A**: 2/3 under 49 released at normal levels. **B**: All under 49 released at normal levels.

If a large fraction of the younger population played no role in transmission of COVID-19, this could actually undermine the results in the present paper, since our findings depend on the ability of the younger population to bear a greater burden of population immunity than the older, at-risk population. (This would not be possible if all observed epidemic growth was actually driven primarily by the older population).

To address the influence of this hypothetical issue, we have modeled additional scenarios where children are 50% less susceptible to infection and/or 90% less likely to transmit once infected. We find that these changes do not have a large enough impact on the transmission dynamics of the epidemic to impact our findings (S5 Fig in S1 File).

Each of these scenarios corresponds to a modified contact matrix; decreased susceptibility for an age group is modeled by scaling the corresponding rows, decreased infectiousness is modeled by scaling the corresponding columns. In Fig 8 we see the resulting contact matrix if we assume that children under 10 are 50% less susceptible to infection, and 90% less infectious once infected. (S2 and S3 Figs in S1 File show analogous plots where individuals under 15 and 20, respectively, have reduced susceptibility and infectiousness). This matrix is then rescaled to correspond to an $R_0$ of 2.7 for the whole population. (Note that scaling this modified contact matrix to $R_0 = 2.7$ requires older age-groups to have higher transmission rates than results

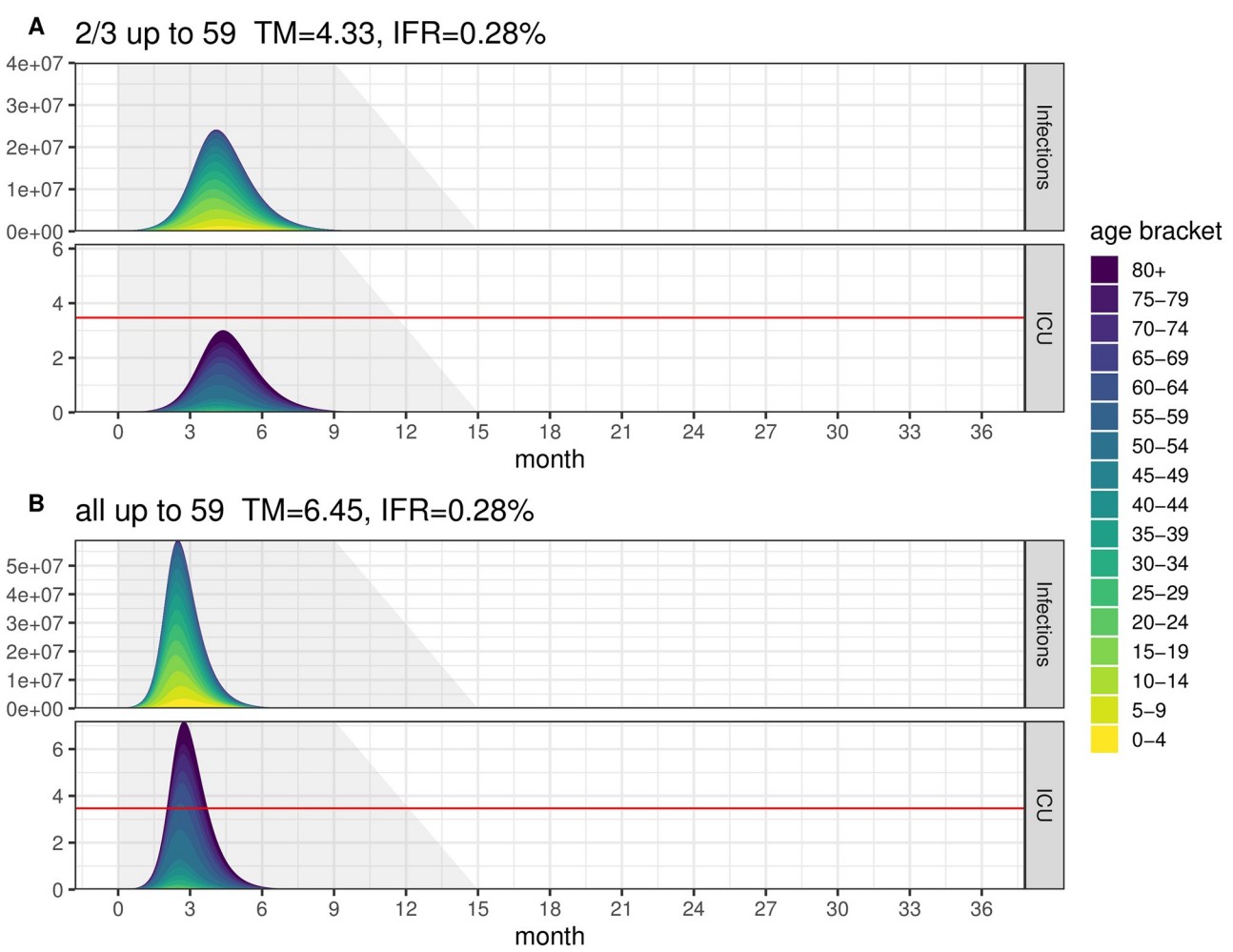

**Fig 6. A**: 2/3 under 59 released at normal levels. **B**: All under 59 released at normal levels.

from rescaling the contact matrix from Fig 1, which is why one might worry that these changes could undermine our findings). The plots on the righthand-side of this figure illustrate that even with these changes, a much smaller number of young age groups suffice to sustain an epidemic than number of older age group which would be required.

As an example, we illustrate the comparison between the use of the default contact matrix as opposed to this modified contact matrix in Fig 9, in the particular case of strategies using age 60 as the cutoff for mitigations. We see relatively small quantitative effects on IFR and total mortality. Other age cutoffs and choices for reduced transmissibility/infectiousness of younger people are discussed in S3 Section in S1 File.

**Remark**: Note that the scenarios in Fig 9 with reduced transmission/susceptibility for children under 10 have a greater IFR, but lower total mortalities. In a qualitative sense, the lower IFR is caused by the greater role older age groups must play in the epidemic under the assumption that children do not contribute to the growth captured by reported $R_0$ values for COVID. The fact that mortalities are nevertheless decreased is caused by the fact that the number of people ever infected is smaller. This is can be understood through the fact that contact patterns are more heterogeneous for the scenario where young children play a lower role (as measured

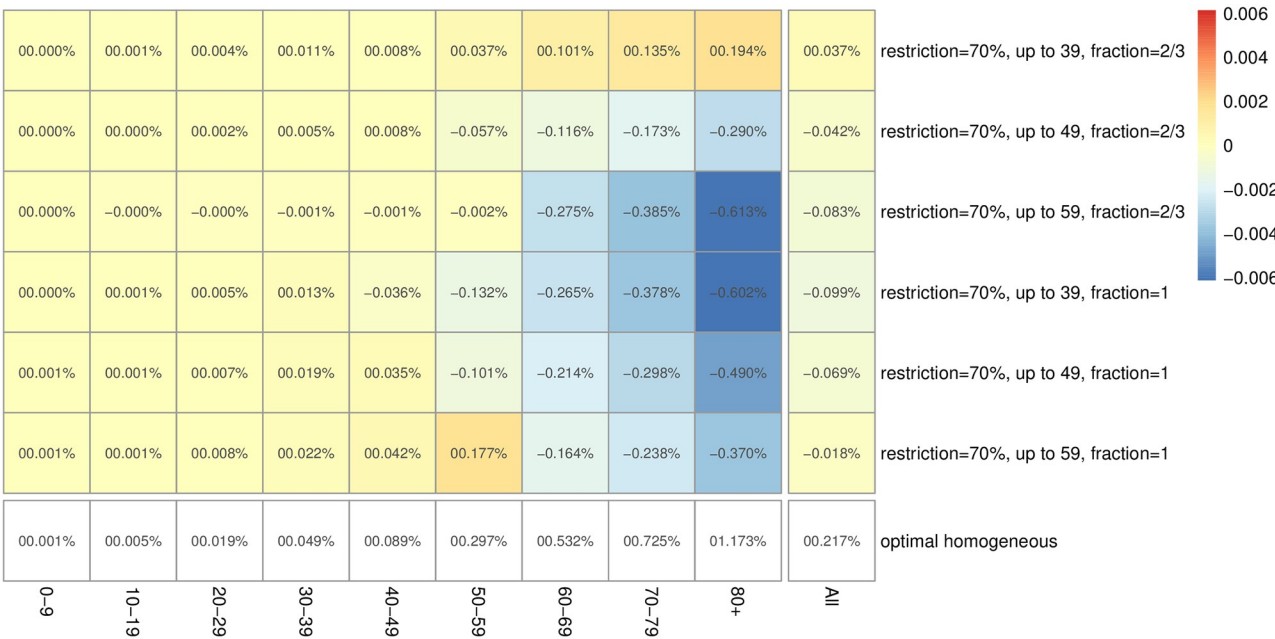

**Fig 7. Impact on mortalities by age group (in groups of 10 years).** The bottom row of this table shows mortalities in each age group for the optimum homogeneous mitigations scenario, as a fraction of the total population of the age group. Note that this is *not* the IFR. Each other row corresponds to a scenario discussed in Section 4.2. The values in the cells in these rows show the *difference* in the mortality rate for each age group for the given scenario, compared with the optimum homogeneous scenario.

by the uniformity of the Peron-Frobenius eigenvector) which leads to the expectation of a lower attack rate. For example, in scenarios where reduced transmissibility and infectiousness apply to everyone under 20, hetereogeneity would decrease and IFR and total mortality would move in the same direction.

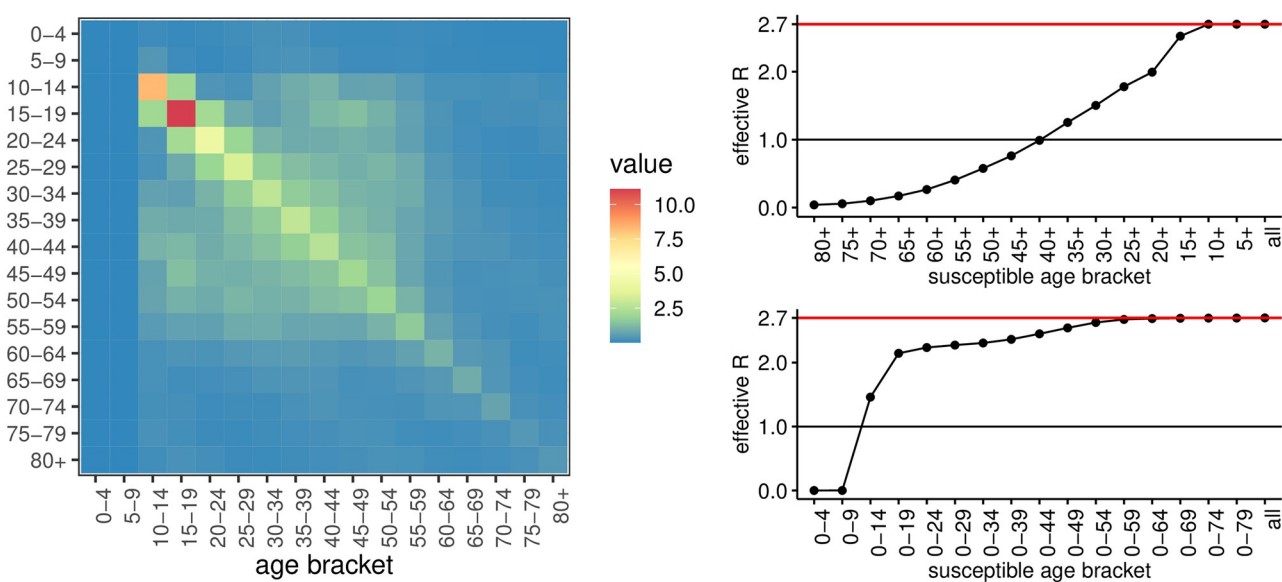

**Fig 8. Left**: A modified contact matrix for the scenario where children under 10 are 50% less susceptible and 90% less infectious than older age groups. **Right**: Even after rescaling to correspond to $R_0 = 2.7$, we see that older age groups are still much less capable of sustaining an epidemic on their own than are younger age groups.

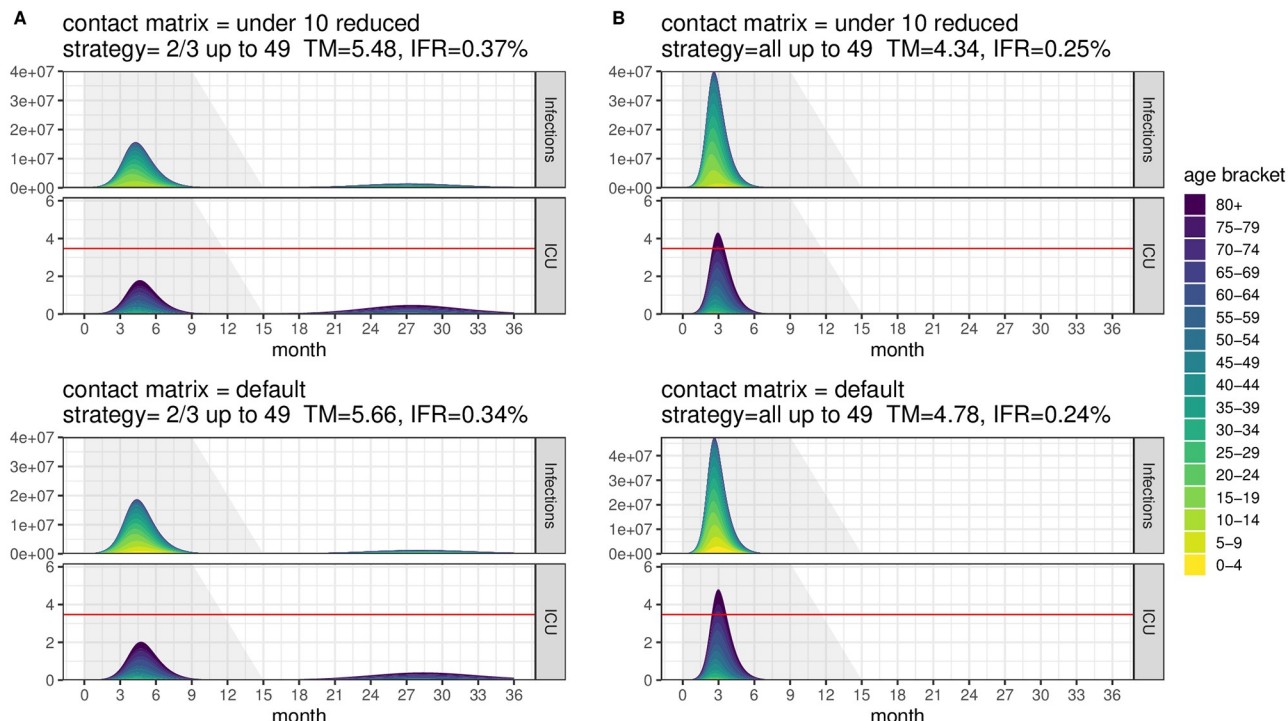

**Fig 9. Comparison of the effect of reduced susceptibility and infectiousness of children under 10.** The top two panels are generated using the modified contact matrix described in Fig 8 as a starting point. The bottom two panels are those generated using the contact matrix from Fig 1 as a starting point; note that these bottom panels are identical to the panels shown in Fig 6. We see only small quantitative differences between the top and bottom panels for each of the two strategies shown on the left and right here (releasing 2/3 or all of the under 60 population, respectively).

## 7 Discussion and caveats

We have considered a model of age-heterogeneous transmission and mitigation in a COVID-19-like epidemic, which is simple but also tied to current estimates of both disease parameters and U.S.-specific contact patterns. We find that age-targeted mitigations can have a dramatic effect both on mortality and ICU utilization. However, we also find that to be successful, age-targeted mitigations may have to be strict. Our scenarios modeling moderate mitigations on the restricted group (shown in S8, S11 and S14 Figs in S1 File) fare quite poorly (see also S2 Table in S1 File).

Importantly, we find that while relatively good strategies exist in a range of scenarios, so long as mitigations on restricted groups can be strict or very strict, the precise choices which minimize ICU utilization and deaths are sensitive—for example, to the fraction of younger people which will actually be released from mitigations.

We also find that if only moderate mitigations are possible on the population subject to mitigations, then the discrete set of age-targeted mitigations we considered fared poorly.

We view our modeling as demonstrating a qualitative point: strict age-targeted mitigations can have a powerful effect on mortality and ICU utilization, even if relative transmission rates among age groups will eventually normalize. We expect that public policy motivated by this kind of finding would have to be responsive; for example, by relaxing restrictions on larger and larger groups conservatively, while monitoring the progress of the epidemic.

Note in S2 Table in S1 File, for every $R_0$ value, there are heterogeneous strategies that outperform the optimal homogeneous strategies. There are also heterogeneous strategies that do worse, sometimes much worse. These poor-performing strategies fall into one of two types:

- Those with too large a relaxed population, so that the initial epidemic is not sufficiently constrained, and

- Those with too small a relaxed population, so that when transmission rates resume, a second wave results (which disproportionately effects the older population).

The simple lessons for policy are twofold: first, **initial relaxations must be responsive to conditions on the ground**, e.g., through monitoring of ICU utilization. Second, **the return to normal must also proceed cautiously** and, ideally, be informed by data on the size of the infected population. Although we have considered simple 2-stage strategies here, it would be natural to consider implementing policies with more stages, as a way of proceeding cautiously.

It is important to emphasize, however, that the extent to which strategies like those considered here are the best approach depends on what other strategies are considered feasible. For example, if the COVID-19 epidemic can be contained indefinitely (i.e., if the second wave in Fig 2B can be avoided) then that approach will certainly prevent more COVID deaths. Our analysis shows, however, that if one believes population immunity will eventually play a role in ending the COVID-19 epidemic in some locations, then age-targeted strategies can have a large effect on the mortality incurred in reaching population immunity.

Any predictive model is an oversimplification of the real world whose predictions depend on parameter values whose true values will only be known after-the-fact. The model we employ is particularly simple, and while this simplicity can be an asset when demonstrating qualitative phenomena, it also presents obvious limitations. For example:

- We do not model seasonality, since the effects of seasonality on COVID-19 remain unsettled. Seasonal forcing could mean, for example, that after relaxing restrictions on some group, they may have to be reinstated as transmission rates increase.

- While we do use (simple) models of known age-group contact patterns, we don't model the effects of specific mitigations on those patterns; for example, we don't evaluate the specific effects of things like closing schools. We also don't attempt to model the different effects of mitigations on within-home and out-of-home contact patterns, since the ways these each contribute to empirically observed $R_0$ values is complicated.

- There is still considerable uncertainty regarding basic parameters of COVID-19 such as its transmissibility, the infection mortality rate, and the ICU admission rate. While we have relied on expert choices for the parameters we have used, all quantitative findings we make (such as ICU utilization) are sensitive to these choices.

- Like prior work modeling the COVID-19 epidemic (e.g., [22], [23]), we do not model nursing home dynamics separately from the broader population. Despite the fact that less than 3% of the U.S. population over 65 lives in nursing homes [24], their elevated contact patterns and large role to date in COVID mortality suggests the need for more work aimed at addressing their role explicitly.

On the other hand, there are ways in which our analysis has been conservative. For example:

- We have only modeled contact heterogeneity in contacts at the age-group level. Further heterogeneous clustering of contacts, or in susceptibility, could further reduce attack rates [25, 26].

- When modeling homogeneous mitigation strategies in Fig 2, we allow ourselves complete freedom to choose an effective $R_0$ value resulting from mitigations. On the other hand, for

our examples of heterogeneous mitigation strategies, we have tied our hands considerably more. We simply allow ourselves to choose which age group, in intervals of 10, to release to normal transmission levels—we have just a few discrete options to choose from. In particular, if were allowed ourselves to combine mild mitigations on the relaxed group with strong mitigations on the rest, we would be able to achieve fewer mortalities and ICU utilization. Note that many of our age-targeted mitigation scenarios exhibit epidemics which end well before the 9-month mark; these curves have room to be flattened.

- We have considered scenarios where not all the younger age group will be able to have relaxed transmission rates, because of cohabitation with older household members, or because of risk factors other than age. This can make our analysis worse, by increasing the average age of the required immunity herd. However, we have not taken advantage of a presumed benefit that this would confer: if effective risk-models incorporating factors beyond age could be deployed, it is plausible that the ICU admission rate for the relaxed groups could be decreased.

## Supporting information

**S1 File.**
(PDF)

**S2 File.**
(TEX)

## Acknowledgments

We have benefited from discussion and helpful comments with regard to this and our previous analysis [27] with many people, including Boris Bukh, Don Burke, Forrest Collman, Jordan Ellenberg, Ryan O'Donnell, Russell Schwartz, Istvan Szapudi among several others.

## Author Contributions

**Conceptualization:** Maria Chikina, Wesley Pegden.

**Data curation:** Maria Chikina, Wesley Pegden.

**Formal analysis:** Maria Chikina, Wesley Pegden.

**Methodology:** Maria Chikina, Wesley Pegden.

**Software:** Maria Chikina, Wesley Pegden.

**Visualization:** Maria Chikina, Wesley Pegden.

**Writing – original draft:** Maria Chikina, Wesley Pegden.

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
