## [Decision Letter · Decision Letter 0]

20 May 2020

PONE-D-20-10135

Modeling strict age-targeted mitigation strategies for COVID-19

PLOS ONE

Dear Dr. Chikina,

Thank you for submitting your manuscript to PLOS ONE. After careful consideration, we feel that it has merit but does not fully meet PLOS ONE’s publication criteria as it currently stands. Therefore, we invite you to submit a revised version of the manuscript that addresses the points raised during the review process.

We would appreciate receiving your revised manuscript by Jul 04 2020 11:59PM. To enhance the reproducibility of your results, we recommend that if applicable you deposit your laboratory protocols in protocols.io, where a protocol can be assigned its own identifier (DOI) such that it can be cited independently in the future. For instructions see: http://journals.plos.org/plosone/s/submission-guidelines#loc-laboratory-protocols

We look forward to receiving your revised manuscript.

Kind regards,

Lidia Adriana Braunstein, Phd in Physics

Academic Editor

PLOS ONE

Journal Requirements:

"The funders had no role in study design, data collection and analysis, decision to

publish, or preparation of the manuscript"

3. We note that a table in your submission may be copyrighted, this is noted with the following text: "The mortality rate and rate of ICU admissions per infection are taken from Report 9 of the team at Imperial College London [4]; we use the data from their Table 3".

All PLOS content is published under the Creative Commons Attribution License (CC BY 4.0), which means that the manuscript, images, and Supporting Information files will be freely available online, and any third party is permitted to access, download, copy, distribute, and use these materials in any way, even commercially, with proper attribution. For more information, see our copyright guidelines: http://journals.plos.org/plosone/s/licenses-and-copyright.

We require you to either (a) present written permission from the copyright holder to publish these table specifically under the CC BY 4.0 license, or (b) remove the table from your submission:

a. You may seek permission from the original copyright holder of Figure(s) [#] to publish the content specifically under the CC BY 4.0 license.

4. Please ensure that you refer to Figure 1 in your text as, if accepted, production will need this reference to link the reader to the figure.

Reviewers' comments:

Reviewer's Responses to Questions

**Comments to the Author**

1. Is the manuscript technically sound, and do the data support the conclusions?

Reviewer #1: No

Reviewer #2: Yes

Reviewer #3: Partly

2. Has the statistical analysis been performed appropriately and rigorously? 

Reviewer #1: N/A

Reviewer #2: Yes

Reviewer #3: N/A

3. Have the authors made all data underlying the findings in their manuscript fully available?

Reviewer #1: Yes

Reviewer #2: Yes

Reviewer #3: Yes

4. Is the manuscript presented in an intelligible fashion and written in standard English?

Reviewer #1: Yes

Reviewer #2: Yes

Reviewer #3: Yes

5. Review Comments to the Author

Reviewer #1: The paper "Modeling strict age-targeted mitigation strategies for COVID-19"

considers an oversimplified, off-the-shelf, SIR model modified by a

social-contact matrix between age-groups. The idea is not new as the authors

let us know.

The novelty of the manuscript would then be limited to its application to the

SARS-CoV-2 pandemic. Hence, it deserves consideration in as much it

incorporates essential particularities of the current problem.

The construction of the model, its relation with the know features of

SARS-CoV-2 does not deserves much attention as the authors rush into the habits

of modelling and their own limited interest.

Nothing worth reading is going to come from ignorance of the phenomena. Even at a qualitative level: which epistemological theorem states that what it is no of my interest will have no influence in the result?

Let me list a few obvious matters that will influence the results but the

authors have not considered.

1. Behavior changes with the illness when symptoms appear.

2. Behavior changes with the social perception of risk in an epidemic.

3. The course of SARS-CoV-2 changes with age. In particular, recovery times.

4. It is suspected that mild cases are less contagious than severe cases

(before isolation).

5. The recovery time that matters is the time from onset of contagiousness to

isolation or end of the contagious period (whatever comes first). Such times

depend on age.

6. The contagious period is not exponentially distributed (without

exponentially distributed times for each compartment, there is no support for

ODE models)

7. Social contact at normal times is not the important kind of contact in terms

of the propagation of the epidemic. What is relevant is the ability to transmit

the illness.

8. A homogeneous contact (without social structure) limits any model to small

communities.

9. An ODE approach limits its scope to large numbers in each compartment.

10. The combination of (6) and (7) may limit the scope to the empty set.

11. R0 is model depending. As such, it cannot be read from the data.

12. Do people in the USA continue with their social-contacts being active

epidemiologically when hospitalized? I would really be surprised. This is just

another feature the authors built in their model without realizing it. It is

the consequence of the faulty epistemology.

The authors, upon giving a fair view to the relevance of the matters they have

ignored may very well decide that they have nothing serious to say. From my

point of view, this work will only be useful to confuse the uneducated.

I did not read beyond section 3, for it makes no sense to consider a toy model

during a period of high demand of serious modelling.

Reviewer #2: In the paper, the authors proposed a SIR-like epidemic model with contact matrix and study the effect of age-targeted mitigation strategies. It is an innovation point of the manuscript. Results are interesting and satisfactory. There are, however, still some minor problems need to be solved before publication.

1. Please give the exact value of contact matrix C when modeling mitigation strategies.

2. Please give the clear description of mitigation strategies.

3. The figures are unclear, especially the figure of ICU.

4. I cannot understand the result of figure 2 B. Why there are no infections at the beginning?

Reviewer #3: In the present work, the authors present a model for a strict age-targeted mitigation strategies for COVID-19. The model is based on a standard SIR model adapted to include an aged specific contact matrix. Also, some age-specific epidemiological parameters were included in the model. The author show how such a strategy can avoid the collapse of the ICU units as the contagion of the elderly is smooth and even lower than in the absence of such a strategy. In turn, the targeted isolation can make the quarantine more tolerable for the rest of the population.

The main results are in part trivial, as a natural result of partially isolation part of a population is preventing them from being infected. In order for this model to prove of some utility would be if it can provide robust qualitative results.

The model uses a contact matrix that is asymmetric due to the methodology used to build it. The results are based in a directed survey, where there is always a pointing and a pointed person. This is the origin of the asymmetry and not because they correspond to frequencies of interactions. It is not clear how the authors build their symmetric matrix. and where they got the information about the population pyramid.

The dynamics of the ICU is not described. There is no accurate information about the permanence of patients in ICU units.

The prevalence of risk groups among the nonisolated population is not taking into account. This information is very relevant at the moment of an accurate estimation of the occupation of ICU.

The information about the percentage of each age group ICU requirement is obtained from data collected from a different country, a different population. A simple research across reports from different countries show how scattered these percentages are.

In the last weeks, we have seen a plethora of models, with a vast majority of them presenting contradicting and out of scale results.

In the present form, the status of the present model is conjectural only, with questionable robustness. It would be irresponsible to propose a public health policy on the basis of such a feeble analysis. The author should present a stronger and more founded model.

6. PLOS authors have the option to publish the peer review history of their article (what does this mean?). If published, this will include your full peer review and any attached files.

Reviewer #1: No

Reviewer #2: No

Reviewer #3: No

---

## [Decision Letter · Decision Letter 1]

6 Jul 2020

Modeling strict age-targeted mitigation strategies for COVID-19

PONE-D-20-10135R1

Dear Dr. Chikina,

We’re pleased to inform you that your manuscript has been judged scientifically suitable for publication and will be formally accepted for publication once it meets all outstanding technical requirements.

Kind regards,

Lidia Adriana Braunstein, Phd in Physics

Academic Editor

PLOS ONE

Reviewers' comments:

Reviewer's Responses to Questions

**Comments to the Author**

1. If the authors have adequately addressed your comments raised in a previous round of review and you feel that this manuscript is now acceptable for publication, you may indicate that here to bypass the “Comments to the Author” section, enter your conflict of interest statement in the “Confidential to Editor” section, and submit your "Accept" recommendation.

Reviewer #2: All comments have been addressed

Reviewer #3: All comments have been addressed

2. Is the manuscript technically sound, and do the data support the conclusions?

Reviewer #2: Yes

Reviewer #3: Yes

3. Has the statistical analysis been performed appropriately and rigorously? 

Reviewer #2: Yes

Reviewer #3: Yes

4. Have the authors made all data underlying the findings in their manuscript fully available?

Reviewer #2: Yes

Reviewer #3: Yes

5. Is the manuscript presented in an intelligible fashion and written in standard English?

Reviewer #2: Yes

Reviewer #3: Yes

6. Review Comments to the Author

Reviewer #2: (No Response)

Reviewer #3: The author have done a correct and thorough revision of their manuscript. I consider that it is ready for acceptance.

7. PLOS authors have the option to publish the peer review history of their article (what does this mean?). If published, this will include your full peer review and any attached files.

Reviewer #2: No

Reviewer #3: **Yes: **Marcelo N Kuperman

---

## [Editor Report · Acceptance letter]

16 Jul 2020

PONE-D-20-10135R1 

Modeling strict age-targeted mitigation strategies for COVID-19 

Dear Dr. Chikina:

I'm pleased to inform you that your manuscript has been deemed suitable for publication in PLOS ONE. Congratulations! Your manuscript is now with our production department. 

Kind regards, 

on behalf of

Dr. Lidia Adriana Braunstein 

Academic Editor

PLOS ONE